# Dwarf Mistletoe Infection Interacts with Tree Growth Rate to Produce Opposing Direct and Indirect Effects on Resin Duct Defenses in Lodgepole Pine

**Scott Ferrenberg** 

Department of Biology, New Mexico State University, Las Cruces, NM 88011, USA; ferrenbe@nmsu.edu

**Abstract:** *Research Highlights:* I sought to disentangle the influences of tree age, growth rate, and dwarf mistletoe infection on resin duct defenses in lodgepole pine, *Pinus contorta* Douglas ex Loudon, revealing the presence of direct positive and indirect negative effects of mistletoe on defenses. *Background and Objectives:* For protection against natural enemies, pines produce and store oleoresin (resin) in 'resin ducts' that occur throughout the tree. Dwarf mistletoe, *Arceuthobium americanum* Nutt. ex Engelm. (hereafter "mistletoe"), is a widespread parasitic plant affecting the pines of western North America. Infection by mistletoe can suppress pine growth and increase the probability of insect attack—possibly due to a reduction in resin duct defenses or in the potency of chemical defenses at higher levels of mistletoe infection, as reported in *Pinus banksiana* Lamb. However, the influence of mistletoe infection on defenses in other pine species remains unclear. I hypothesized that mistletoe infection would induce greater resin duct defenses in *P. contorta* while simultaneously suppressing annual growth, which was expected to reduce defenses. *Materials and Methods:* Using increment cores from *P. contorta* trees occurring in a subalpine forest of Colorado, USA, I quantified tree age, annual growth, annual resin duct production (#/annual ring), and cross-sectional area (mm$^2$ of resin ducts/annual ring). *Results:* Mistletoe infection increased with tree age and had a direct positive relationship with resin duct defenses. However, mistletoe infection also had an indirect negative influence on defenses via the suppression of annual growth. *Conclusions:* Through the combined direct and indirect effects, mistletoe infection had a net positive impact on resin duct production but a net negative impact on the total resin duct area. This finding highlights the complexity of pine defense responses to natural enemies and that future work is needed to understand how these responses influence overall levels of resistance and the risk of mortality.

**Keywords:** conifer; natural enemies; oleoresin; parasitic plant; pines; plant defense; tree age

## 1. Introduction

Over their lifespans, trees interact with an array of natural enemies. For defense, conifers utilize oleoresin (hereafter "resin") composed of various terpene compounds [1–3]. Resin can be directly toxic to natural enemies and act as a mechanical deterrent that exudes from wounds to mire attackers and seal damaged tissue [1–3]. In many genera of conifers, resin is produced and stored in specialized cells called "resin ducts" [1,2]. As an indication of their importance for tree resistance to natural enemies, during bark beetle outbreaks in multiple species of pines, greater numbers and cross-sectional areas of vertical resin ducts within the xylem have been shown to increase the likelihood of a tree avoiding [4] and surviving an attack [5,6]. Because resin defense phenotypes are heritable [7–9], attacks by natural enemies should exert directional selection that increases the frequency of 'well-defended' phenotypes within populations. Despite the potential increase in fitness associated with producing more or larger

resin ducts, these defenses can vary substantially within and among populations [10]. Understanding what drives this variation remains central to predicting the susceptibility of pines to natural enemies.

Numerous factors have been implicated as drivers of variation in pine defenses. For example, defenses have been demonstrated to vary across gradients in resources [8] or climate [11–13] and among stages of ontogeny [14,15] and reproduction [16]. Tree growth rate and size have also been found to influence resin defenses, with slower growth rates correlated to a reduced production of resin ducts [17] and lower chemical defenses [12]. Another potential, yet less-studied driver of variation in resin duct defenses is infection by pathogens or parasites that can induce systemic, long-term changes in defenses [18–20].

In pines, the parasitic plants known as dwarf mistletoes, *Arceuthobium* spp. (hereafter simplified to "mistletoe"), are widespread agents of infection in forests of the intermountain region of western North America (Figure 1A). Mistletoe attaches to a host tree and subsequently extracts water and nutrients. As a consequence, mistletoe infection can reduce rates of photosynthesis and tree growth [21,22] and increase rates of attack by bark beetles [23,24] and mortality across a range of conifer species [25–33]. The increase in bark beetle attack and mortality correlated to mistletoe infections could be due to a reduction in tree growth and resin duct defenses, or in the potency of chemical defenses at higher levels of infections, as reported in *Pinus banksiana* Lamb. [34]. However, the influence of mistletoe infection on growth and defenses in other pine species has revealed potentially conflicting responses among species as infections increased growth rates in *Pinus ponderosa* Lawson and C. Lawson and *Pinus aristata* Engelm. [24,35].

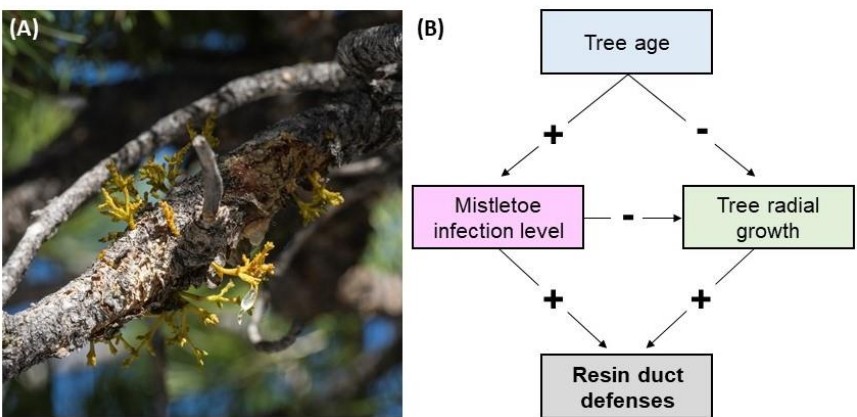

**Figure 1.** (**A**) Dwarf mistletoe infection causes swelling on a small brank of a lodgepole pine (*Pinus contorta*) (photo by Jeffry Mitton); (**B**) the hypothesized model of resin duct defenses as a function of mistletoe infection, tree age, and tree growth rate.

Across Colorado, USA, the mistletoe *Arceuthobim americanum* Nutt. ex Engelm. (commonly known as "lodgepole pine dwarf mistletoe") infects up to 50% of *P. contorta* (subsection *latifolia*) in montane and subalpine forests [36]. Tree age has been shown to be an important factor that positively correlates to the likelihood of mistletoe infection in some conifers [26,37]. Because resin defenses have also been reported to be influence by tree age, size, and growth rate [12,17], additional efforts to disentangle the relationships among these tree properties and mistletoe infection are needed. Thus, quantifying the effects of mistletoe infection on tree defenses, particularly resin duct traits previously shown to influence survival of *P. contorta* during bark beetle attack [5], is an important goal for elucidating the roles of various interacting drivers of tree mortality in the face of global change pressures. To support this goal, I tested three predictions regarding direct interactions among tree age, growth rate, mistletoe infection level and probability, and resin duct defenses: (1) resin duct defenses would be positively related to tree growth; (2) tree growth rate would decline with age; and (3) the level (severity) and probability of mistletoe infection would increase with tree age, most likely as a result of increased exposure to the parasite.

Given the above predictions and recent work in *P. banksiana*—a closely related congener of *P. contorta*—that demonstrated that mistletoe can have both positive and negative effects on defenses [34], I posited a set of direct and indirect interactions among mistletoe infection and tree growth, defenses, and age. Specifically, I hypothesized that mistletoe infection would have a direct positive effect on resin duct defenses due to trees inducing defenses in response to infection but would simultaneously have an indirect negative-effect on resin duct defenses via the suppression of tree growth. Meanwhile, tree age would enhance this negative effect given the predicted decline in tree growth rate and increase in mistletoe infection level. This a priori hypothetical model (Figure 1B), specifying the relationships among tree properties, infection, and resin duct defenses was subsequently tested using data collected from *P. contorta* of the Colorado Rocky Mountains, USA.

## 2. Materials and Methods

### 2.1. Study Site and Data Collection

The tree data used in this study were collected in the summer of 2013 at the University of Colorado Mountain Research Station in Boulder County, CO, USA. This site was centered at 40.0294, −105.5293, which is located 2845 m above sea level on the eastern slope of the Colorado Front Range of the US Rocky Mountains. This site is a relatively flat bench with an east to south-east aspect. The climate and soils of this study site were described by Duhl, et al. [38] and long-term climate data indicate that >1.6 °C of warming has occurred in this subalpine forest since 1970 [39]. Bark beetle outbreaks affected numerous overstory trees of this study area between 2008 and 2013. Prior work with bark beetle host selection indicates a preference for attacking less-defended trees in terms of resin duct production; thus, I designated a four hectare area composed of relatively young trees, with no visible bark beetle activity or recent mortality, within which the sampling was completed. Sampled trees were selected from along a randomly oriented transect initiated 50 m from the edge of a public access road and terminated after 250 m. Sampling began by selecting the first *P. contorta* tree whose canopy intersected the 250 m transect, and then sampling the next available *P. contorta* that had both a canopy overlapping the transect and was a minimum of 10 m from the previously sampled tree. Once a tree was selected, I collected one 12 mm diameter increment core at 1.37 m above the ground surface and mistletoe infection level was quantified by two pairs of observers using the 6-point Hawksworth rating system [40]; the two sets of infection data were then averaged to derive each tree's rating—an approach that allows for fractional values within the traditional 6-point scale. Mistletoe infection rates are relatively high among *P. contorta* in this location [28,36]. The mean Hawksworth rating for sampled trees was 2.67; 15 of the 21 randomly selected trees had a rating >0 and 10 trees had a Hawksworth rating ≥3.

Increment cores were air-dried for one week, mounted onto a wooden block, and sanded with a progression of coarse to fine sandpaper to create a flat cross-section for analysis [5]. I then visually cross-dated each increment core, through the period from 2003 to 2012, which corresponds to the time series over which resin duct metrics were assessed. After scanning each increment core at 4800 dpi using an Epson V750 flat-bed scanner, I used ImageJ within the FIJI platform [41] to measure the annual radial growth (mm per year) for each visible ring (from the pith through 2012) and the cross-sectional resin duct area ($mm^2$ of each duct) for all resin ducts in each annual growth ring between 2003 and 2012 (i.e., 10 years of annual growth). Importantly, while increment cores were collected in 2013, tree ring measures associated with 2013 were excluded from subsequent calculations and analyses because secondary growth and resin duct differentiation was unlikely to have been completed at the time of the summer sampling.

I estimated the age of each tree by counting all visible growth rings in each increment core. While missing rings are possible in these trees, missing rings are rare in *P. contorta* growing in cool environments—like the subalpine study site used here—relative to those growing in warmer environments [42]. I assessed the accuracy of cross dating with the "dplR" package for R [43]. Without

applying a detrending procedure, the series inter-correlation of this period was 0.53 with no missing annual rings suggested by lagged-correlation analysis. In addition to cross dating, I used annual radial growth measures to quantify each tree's annually resolved diameter by assuming a circular structure of the stem and doubling the tree's radius and to calculate annual growth as a basal area increment (BAI), which standardized annual radial growth as the proportion of total cross-sectional area each ring represented for a given tree. I also used annual radial growth to calculate an annually corrected diameter at breast height (DBH; 1.37 m above the ground surface) for each tree and combined this data with a published allometric scaling equation to calculate the aboveground biomass change for each tree using the approach described in Stephenson et al. [44]. Following the removal of the 2013 data, mean tree age in this study was 48 years with a range from 15 to 98 years, while mean DBH was 60.04 mm with a range of 26.99 to 118.00 mm.

*2.2. Data Analyses*

For all analyses and data manipulation, I used R version 3.4.3 [45]. I used linear mixed models in the "lme4" package [46] to assess (1) the relationship among the number and total cross-sectional area of resin ducts produced in annual growth rings (Section 3.1), as well as the validity of prediction one regarding resin duct number/the total area and tree growth and age (Section 3.2) and the validity of prediction two regarding tree radial growth/BAI growth and tree age and size (Section 3.3). All mixed models included the intercept for tree identity as a random effect to account for the nested structure of the annually resolved growth and defenses data and to avoid pseudo-replication. To aid in assessing the fit of mixed models to the data, I computed conditional and marginal $R^2$ values for each model using the "MuMIn" package [47]. For all models, data transformations sufficiently improved the distribution of the response variables to allow models to be tested with a Gaussian distribution (see Table S1 for data transformation descriptions).

To assess my third prediction regarding the influence of tree age on mistletoe infection status (Section 3.4), I used the "lmPerm" [48] package to complete a permutational ANOVA testing the importance of tree age for mistletoe infection level followed by a logistic regression classifying trees as infected vs. uninfected using tree age as a predictor. I assessed the logistic model fit by determining the proportion of trees properly classified as infected (1) or uninfected (0) and by the McKelvey-Zavoina pseudo-$R^2$ [49]—calculated using the "BaylorEdPsych" package [50]—as recommended by Windmeijer [51] as the best ordinary least squares (OLS) estimator in logistic regression.

Finally, to test the hypothesized a priori model (Figure 1B), I used the "lavaan" package [52] to fit structural equation models (SEMs) assessing tree property influences on (1) resin duct numbers and (2) total cross-sectional area (Section 3.5). SEMs were fit to binned growth and defense data from the most recent five years of annual rings (i.e., 2008 to 2012) using maximum likelihood estimation with robust standard errors (i.e., estimator = "MLM"). Unlike the annually resolved linear mixed models described above, multiple years of data were binned for the SEMs given a lack of knowledge regarding the dates when trees suffered initial mistletoe infection and rates of increase in infections over time. As a result, mistletoe ratings for trees were not annually resolved through time in the same manner as growth and defense. I selected a five-year bin for two reasons: first, this window of time is long enough to reveal potential mistletoe impacts on tree growth and defenses but short enough to help reduce the overestimation of past infection levels, and, second, the binning of the most recent five years of tree growth and resin duct data is a common practice that has been employed in many studies of pine defenses, thereby making the results easier to compare to the available literature (e.g., [4–6])

## 3. Results and Discussion

### 3.1. Relationship of Number and Area of Resin Ducts

The total cross-sectional areas of resin ducts per annual xylem ring positively related to total resin duct counts per annual ring ($p < 0.0001$; Figure S1). The mixed model, using tree identity as a random effect, had a marginal $R^2 = 0.82$ (variation in total resin duct area per ring explained by the number of resin ducts per ring) and a conditional $R^2 = 0.93$ (variation collectively explained by the fixed and random effect—i.e., the tree identity). This result confirms that total resin duct area is well predicted by resin duct counts, highlighting the potential to use this less time-intensive measure in studies of pine defenses when such a relationship can be verified. Previous work in pines also found a significant relationship among resin duct area and counts [4,6], but in these and other reported efforts, resin duct metrics were derived for binned groups of annual rings (e.g., 5 or 10 years of annual rings collectively) and were not analyzed using annualized data. Despite the strong correlation among these measures, the influence of the random effect for improving the fit of models highlights the value of continued characterization of what determines allocation to different aspects of resin duct defenses—i.e., more versus larger resin ducts—within and among individuals of a population.

### 3.2. Prediction 1: Resin Duct Defenses and Tree Growth

Both measures of resin duct defenses—i.e., the number and total area of resin ducts in annual rings—were significantly related to rates of annual radial growth ($p < 0.05$) (Table S2) when tree age and size were accounted for in mixed models. The mixed model for the number of resin ducts per annual ring had a marginal $R^2 = 0.34$, a conditional $R^2 = 0.40$ (Figure 2A). For total resin duct area, the mixed model had a marginal $R^2 = 0.38$, a conditional $R^2 = 0.48$ (Figure 2B). A positive relationship between tree growth and resin duct defenses has been reported for several species of pines [4,6,9,16,34], including for *P. contorta* sampled within the site used in the present study [5]. This finding is also supported by a meta-analysis that found that terpenoid-based defenses tend to have a positive relationship to growth and only exhibit negative relationships when resources are highly abundant [53].

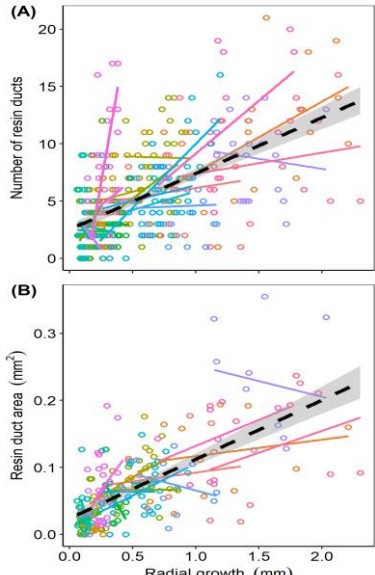

**Figure 2.** The relationship of annual resin duct counts (**A**) and cross-sectional resin duct area (mm$^2$ per year) (**B**) to annual radial growth (mm per year). Both panels show untransformed data. Colored symbols and solid lines correspond to individual trees, the larger dashed line shows the population line of fit with shading indicating the 95% confidence interval.

A positive relationship among tree growth and the production of resin defenses—including resin duct and chemical measures—has been reported across numerous species of pines [12,17]. This common result counters predictions made by the growth–differentiation balance hypothesis (GDBH) of plant defenses, which is oft-invoked in studies of pine resin defenses [17]. This growing refutation of the GDBH in studies of pine defenses suggests that this hypothesis is either poorly suited to this group of plants under typical environmental conditions, or it is not well-suited to studies that focus on defenses produced in the stem of pines in the absence of information on allocation to growth and defense in other compartments of a tree (i.e., roots, limbs, needles, and cones). Importantly, a trade-off among resin duct defenses and cone production has been reported in *Pinus edulis* Engelm. [16], suggesting possible trade-offs beyond growth and defense in conifers. Regardless, the GDBH seems to function primarily as a framing tool in pine-defense studies and not as a mechanistic prediction model that is subsequently or robustly tested. Future efforts to characterize stem growth and defense within conifers should acknowledge the apparent weakness of the GDBH and consider either alternative experimental designs for testing the GDBH, or alternative hypotheses within the larger body of plant defense theory [54].

### 3.3. Prediction 2: Influence of Tree Age on Growth

Tree growth rate, measured as annual radial growth, was negatively related to tree age ($p < 0.0001$) and positively related to tree size (DBH) ($p > 0.05$), which was included in the initial model to account for potential changes in radial growth due to tree diameter (Table S3, Figure 3A). The mixed-modeling for radial growth had a marginal $R^2 = 0.69$ and a conditional $R^2 = 0.91$. Tree growth was also significantly and negatively related to tree age when measured as annual BAI ($p < 0.0001$; Table S3, Figure 3B), which accounts for changes in stem diameter by measuring growth rate in proportion to overall tree size. The mixed model for BAI as a function of tree age had a marginal $R^2 = 0.83$ and a conditional $R^2 = 0.95$. A decrease in radial growth relative to tree age, was previously reported for *P. contorta* populations sampled from within and near my study area [12]. Initially, this result appears to conflict with the reported positive relationship among age and annual, aboveground-biomass accumulation found across a wide-range of tree species, including *P. contorta* of the western US [44]. However, after applying the allometric equation used by Stephenson et al. [44] to scale tree diameter to aboveground biomass for *P. contorta*, a relatively small but positive relationship among these measures was revealed (coefficient = 0.0197, $p < 0.0001$; Figure S2). This result generally supports a metabolic scaling theory prediction that mass growth rate should increase continuously with tree size [44,55]. However, older trees from this population exhibited either non-significant relationships among mass growth and age, or a small decline in mass growth (Figure S2)—a result that aligns with numerous and often contentious reports of age-related decline in forest productivity [56]. While this data suggests a shift in the mass growth–age relationship over time, this topic was not the focus of the present study, and the sample was not of sufficient size to question the well-replicated, multi-species analysis of Stephenson et al. [44]. Additionally, work in various species of conifers has indicated that mistletoe infection can alter the accumulation and distribution of biomass across the regions of a tree (i.e., changes to stem vs. leaf vs. limb/branch biomass ratios) [57]. This influence of mistletoe on biomass distribution and the rate of accumulation is certain to affect allometric relationships within and among trees, introducing some uncertainty regarding biomass calculations. Efforts to understand the effect of mistletoe on aboveground biomass in this *P. contorta* population and how this effect might influence age-related growth rates requires further study.

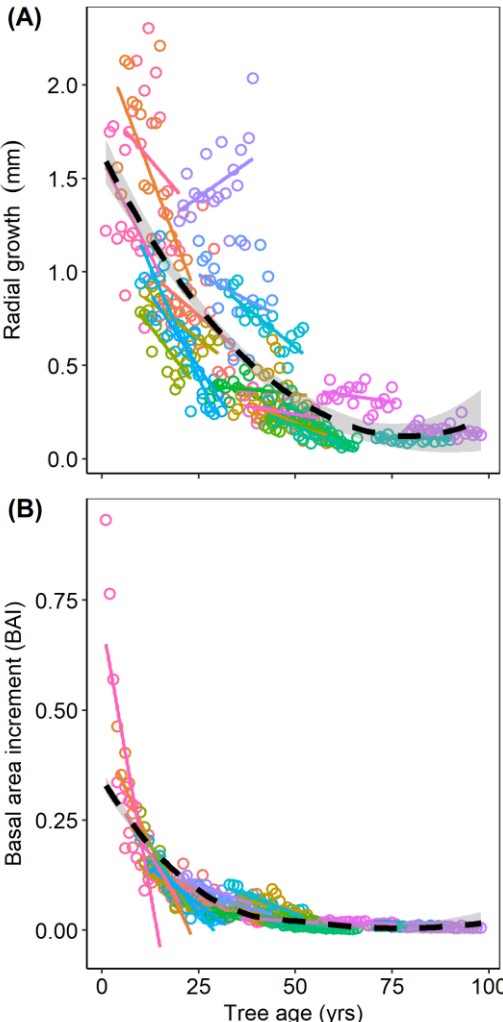

**Figure 3.** Annual radial growth (mm per year) (**A**) and annual basal-area increment growth (BAI; the proportion of a tree's total cross-sectional area to date in mm$^2$ represented by annual ring area in mm$^2$) (**B**) as a function of tree age at the time of annual ring production. Different colored symbols and solid lines correspond to individual trees. The dashed line shows the population line of fit (using loess smoothing), with shading indicating the 95% confidence interval. The panels show untransformed data.

### 3.4. Prediction 3: Influence of Tree Age on Mistletoe Infection

When considered via multiple regression, the mistletoe infection level was significantly related to tree age ($p = 0.0176$) but not to tree size (DBH) ($p > 0.05$), with the model having an $R^2 = 0.22$ (Figure 4). Tree age was also a significant predictor of a tree's probability of being infected by mistletoe; a logistic regression correctly classified 80.9% of trees as infected or uninfected based on the sole predictor of tree age (pseudo-$R^2 = 0.55$, $p = 0.0384$; Figure 4). The positive association between tree age and mistletoe infection is most likely due to each tree having an increased exposure to infection as a function of time. Tree age has previously been reported to positively correlate to the severity of mistletoe infection in conifers [26,33,37]. However, work in *P. aristata* has revealed an exception whereby smaller, younger trees were more likely to be infected than older, larger trees in northern Arizona, USA [24]. These conflicting results for mistletoe infection associations with tree age suggest contexts where age lags behind other tree attributes, such as landscape position or phenotype in determining infection risk, or where forest stand characteristics, such as tree density and species diversity, are a primary determinant of mistletoe spread [58]. Landscape factors, such as forest arrangement, fragmentation level, and climate, as well as stand-level factors, such as tree density, have all been reported to influence mistletoe

infection levels [59,60]. For example, a situation where recruitment occurs within gaps or beneath an existing pine canopy could place younger trees in close proximity to heavily infected trees of a greater age, thereby altering the strength and significance of any age-infection relationship across populations.

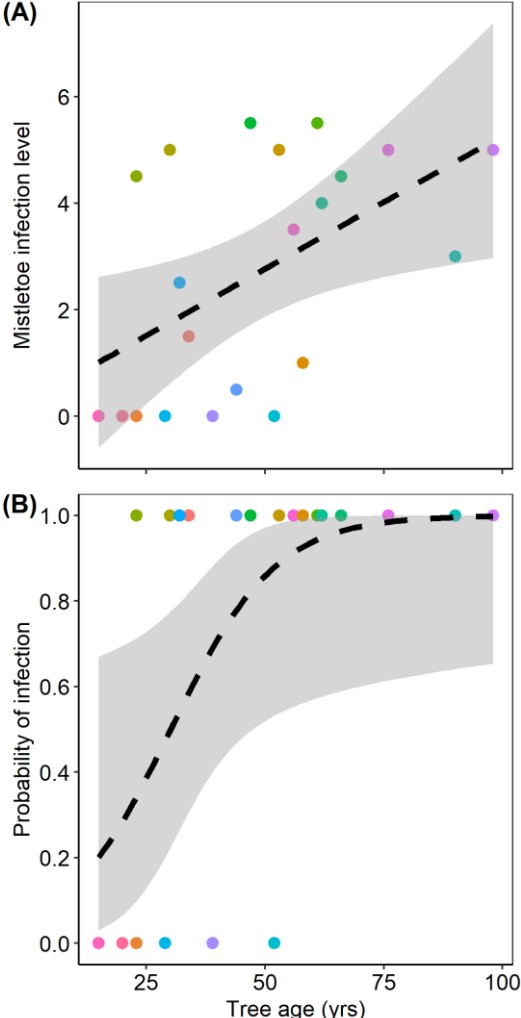

**Figure 4.** Relationship between tree age (total annual xylem rings) and mistletoe infection level (**A**) and the probability of mistletoe infection (**B**). The different colored symbols correspond to individual trees; the dashed-line shows the population line of fit, with shading indicating the 95% confidence interval. The panels show untransformed data.

### 3.5. Hypothesized Direct and Indirect Effects of Mistletoe on Resin Duct Defenses

As reported above, tree age had a significant effect on mistletoe infection level and tree growth, while tree growth had a significant effect on resin duct defenses. Given these responses, determining the overall influence of mistletoe infection on resin duct defenses required fitting a model that accounts for both direct and indirect effect pathways (i.e., Figure 1B). To quantify these pathways, I used structural equation models (SEMs) to parse the direct effect of mistletoe infection on resin duct number and cross-sectional area in the five most recent annual rings of each *P. contorta* tree, as well as the indirect effect mediated by mistletoe impacts on tree growth rate. Both SEM models were considered to be well fit to the data based on the Satorra–Bentler $\chi^2$ value of 0.02 and $p = 0.90$ for the resin duct number and a Satorra–Bentler $\chi^2$ value of 0.75 and $p = 0.39$ for the total resin duct area (Note: small values of $\chi^2$ and large $p$-values indicate SEMs that are well fit to observed data). These SEMs—displayed with standardized coefficients to allow comparisons among the influence of each

illustrated pathway—confirmed the significant positive effect of tree age on mistletoe infection and the negative effect on tree growth rate, while also revealing a significant negative effect of mistletoe infection on tree growth rate (Figure 5). With regard to both metrics of resin duct defenses—i.e., the number of resin ducts produced and the total cross-sectional area ($mm^2$) of resin ducts—mistletoe infection had a direct positive effect (likely by causing trees to induce defenses) and an indirect negative effect via the suppression of tree growth rate (Figure 5). Given the relative strengths of the direct and indirect pathways, the SEM for the number of resin ducts revealed that mistletoe infection had a net positive effect on defenses in the recent annual growth of *P. contorta* (i.e., 0.39 vs. −0.31; Figure 5A), while the SEM for the total area of resin ducts revealed a small, net negative effect (i.e., 0.31 vs. −0.32; Figure 5B). Collectively, these results indicate that mistletoe infection increases the production of resin ducts but appears to simultaneously drive a reduction in their size leading to a net loss of total area (or to a cancelling—in terms of total area—of the potentially positive effect mistletoe infection could have on total resin duct area given the increase in resin duct numbers).

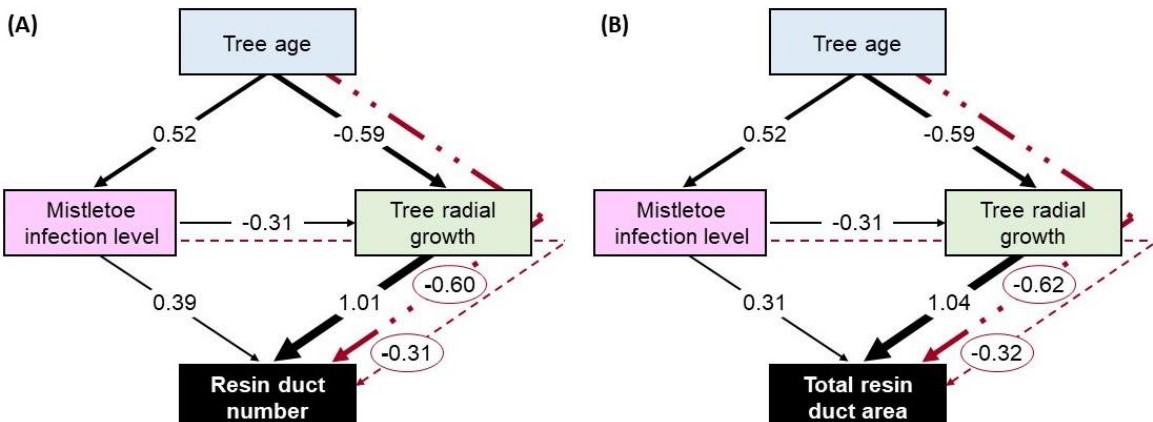

**Figure 5.** Structural equation model (SEM) results illustrating the direct (solid black lines) and indirect (dashed red line) effects of mistletoe infection, tree growth, and tree age on resin duct numbers (panel A) and the total cross-sectional area ($mm^2$) of resin ducts (panel B) within the five most-recent annual xylem-rings of lodgepole pines. The numbers are standardized coefficient values and the arrow sizes are scaled to support comparisons of relative influences among variables. Increasing levels of mistletoe infection have a positive influence on resin duct number and area that is countered by an indirect negative effect mediated by impacts of mistletoe infection on tree growth. All illustrated paths are considered significant ($p < 0.05$).

Prior work across a range of plant species and functional types indicates that defenses can increase or decrease across stages of ontogeny [14,15]. In this population of *P. contorta* trees, age had an indirect negative effect on resin duct defenses—measured either as the number of ducts produced or the total cross-sectional area—that was mediated via an age-related decline in tree growth rate (Figure 5). Prior work with chemical defenses in *P. contorta* stems also revealed an age-related decline in tree defenses [61]. These findings agree with evidence that during recent bark beetle outbreaks in western North America, larger diameter pines (a correlate of age) suffered more bark beetle attacks per surface area of bark [62]. Nevertheless, several studies have reported that chemical defenses—particularly terpenoid compounds—increased with age in various conifer species [15,63,64], including monoterpene concentrations in the xylem tissue in stems of *P. contorta* [12]. Further work is needed before age-related patterns of pine defenses can be fully appreciated.

The indirect negative effect of mistletoe infection on resin duct defenses, which results from the suppression of tree growth by mistletoe infection, adds a mechanistic explanation to prior reports of lower growth rates and higher mortality linked to mistletoe infection level [21,22,25–30,32,33]. Given that bark beetles are more likely to attack and kill trees with fewer or smaller resin ducts [4–6], the suppression of resin duct sizes and total area by mistletoe infection could help to explain the higher

rates of attack by bark beetles on pines with severe mistletoe infections [23,24]. Work in *P. contorta*, has shown that resin duct size and production (numbers) are both important predictors of resistance [5,65]. Nevertheless, trees that make fewer but larger resin ducts appear to have a higher survival probability compared to trees producing more but smaller resin ducts within a population [65]—a result that suggests the existence of an important threshold among resin duct number and size for the effectiveness of defenses against bark beetles. Additionally, increasing levels of mistletoe infection in *P. banksiana* were reported to decrease the concentration of defensive chemicals in resin, which could explain some or all of the change in the risk of insect attack and mortality in infected versus uninfected trees [34]. Whether a change in both the anatomical and chemical defenses of *P. contorta* occurs in response to mistletoe infection will require further exploration. At the same time, the indirect influence of mistletoe on pine defenses, as mediated by impacts on annual growth rate, do not appear to be easily summarized across pine species, as mistletoe infection was reported to increase growth rates in *P. ponderosa* and *P. aristata* in some locations [24,35].

## 4. Conclusions

While the role of resin in host resistance to various natural enemies in isolation has been established, in many cases, a pine tree can be simultaneously confronted by multiple species of enemies, necessitating concurrent responses to avoid or withstand these pressures. How these varied sets of pairwise interactions might combine to affect functions such as growth and defense and ultimately shape a tree's risk of mortality remains enigmatic for most coniferous species. At the same time, recent studies in pines have revealed that levels of chemical and resin duct defenses are influenced by multiple drivers, including tree age, size, growth rate, reproductive output, and landscape position along climatic gradients [12,16,17]. These findings highlight the value in an approach aimed at disentangling a suite of interacting drivers of resin duct defenses in pines. In the present study, this approach revealed a direct positive effect of mistletoe infection on resin duct defenses in annual rings, as well as an age-related increase in the levels of mistletoe infection and an age-related decrease in rates of radial and BAI growth. Despite the increase in resin duct defenses induced by mistletoe infection, the simultaneously occurring age-related changes act to reduce the total cross-sectional area of resin ducts in *P. contorta*. This cumulative outcome occurs via a reduction in resin duct sizes that counters the observed increase in resin duct numbers driven by mistletoe infection. While seemingly minor, a loss of the total cross-sectional area of resin ducts as mistletoe infection level increases could have serious consequences for tree resistance to natural enemies, particularly primary bark beetles. For instance, multiple studies comparing the resin duct properties of trees that survived versus trees that died during bark beetle outbreaks have indicated a positive influence of resin duct numbers and sizes on tree resistance [17]. Yet, evidence from comparisons of *P. contorta* killed by bark beetles versus those that survived suggests that investing in larger resin ducts is a better defensive strategy for this species than increasing the number of resin ducts [65]. Thus, my findings offer a potential mechanism underlying prior reports of increasing rates of bark beetle attack and pest-related mortality resulting from mistletoe infection (e.g., [24,30,32,33]). My results also highlight the need for additional studies, similar to that of Klutsch and Erbilgin [34], that consider multiple measures of resin defense in pines infected by mistletoe and then work to unravel the various direct and indirect pathways through which abiotic and biotic factors shape these defenses in conifers.

**Supplementary Materials:** The following are available online at http://www.mdpi.com/1999-4907/11/2/222/s1, Figure S1: Relationship among total resin duct counts and total cross-sectional area (mm$^2$) of resin ducts, Figure S2: Annual aboveground biomass growth rate (kg/year), Figure S3: Radial growth (mm per year) of the five most-recent annual rings as a function of mistletoe infection level, Table S1: Specified linear models, data transformations used, and analytical approach employed, Table S2: Linear mixed-model for the influence of tree properties on the number and total area of resin ducts in annual growth-rings, Table S3: Linear mixed-model for the influence of tree properties on annual radial growth and annual basal area increment (BAI), Table S4: Permutational ANOVA results testing for the influence of tree age on mistletoe infection level, Table S5: Logistic regression results linking tree age to mistletoe infection category (y/n), Table S6: Structural equation model results.

**Funding:** This research was funded by the Indian Peaks Wilderness Alliance and the John Marr Fund for alpine plant ecology.

**Acknowledgments:** I thank the high school student volunteers from Boulder Valley School District who assisted with tree increment coring and mistletoe infection ratings; and The University of Colorado's Mountain Research Station, Bill Bowman and Jeff Mitton for providing photos and logistical support. I thank Steven Lee and Carla Vásquez-González for comments that aided in crafting this manuscript.

**Conflicts of Interest:** The author declares no conflict of interest. The sponsors had no role in the design, execution, interpretation, or writing of the study.

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
