# Peer review of "Dwarf Mistletoe Infection Interacts with Tree Growth Rate to Produce Opposing Direct and Indirect Effects on Resin Duct Defenses in Lodgepole Pine"

_forests, doi:10.3390/f11020222_

Round 1

Reviewer 1 Report

The paper describes the relationship between tree age, tree radial growth, mistletoe infection level and resin duct defences in lodgepole pine in order to understand better how pine defense responses to natural enemies influence tree resistance and vigour. This work is interesting and relevant, studying pairwise-interactions related to growth and defense by several models and establishing direct and indirect effects.

The paper is generally well-written, however some improvements are needed to clarify and makes easier to the reader the models description in material and methods, results and supplementary tables and the relationships between the factors studied.

Material and Methods:

2.1. Study site and collection:

Please give more detail on mistletoe infection level of the 21 trees selected according to the 6-point Hawsksworth rating system, eg. Number of trees in each rate. In figure S3, from infection level 2 to higher levels, the curve of radial growth is almost flat, how this could be explained?

Also more detail on the tree age of selected trees, according to Figure S2, there seems to be a high variation in age of tested trees. In figure 4 trees range from 5 to 100 years old. The influence of tree age on resin duct number is close to significant P=0.05. The conclusions in Figure 5 of all the models don’t include the relationship between tree age and resin duct number. This aspect may have an effect in your conclusions “a direct-positive effect of mistletoe infection on resin duct defenses in annual rings, as well as an age-related increase in the levels of mistletoe infection”

2.2. Data analysis:

Reading and understanding this section is not easy. Break the very long paragraph in several paragraphs. Description of models should be done by using a “name” and/or “code” per model. This should be the same in material and methods, results and Table S1 by adding a new column in this table. Also replace in this table the word “Section” by “Results Section”.

Tables S2 and S3 are a description of models for section 3.1 and 3.2 models of Table S1. I miss description of models for 3.3, 3.4 and 3.5, so add tables for these models.

Results. Change this heading to “Results and Discussion”

The text just before Figure 4: “these results indicate that mistletoe infection increases the production of resin ducts, but appears to simultaneously drives a reduction in their size leading to a net loss of total area”. This is not clear to me since in Figure 4 values for model A (resin duct number) and model B (total resin duct area) are of similar magnitude and sign (positive or negative)

Reviewer 2 Report

The author of the manuscript titled “Dwarf-mistletoe infection interacts with tree growth rate to produce opposing direct- and indirect-effects on resin duct defenses in lodgepole pine” analyzes the relationships among age, growth, anatomical defenses and dwarf mistletoe. Resin ducts were found to be positively related to dwarf mistletoe, but there was a net negative effect of dwarf mistletoe on resin duct area via the reduction of radial growth. Some growth-defense relationships is discussed in relation to GDBH and how it may not be a fit to pine growth-defense relationships.

A brief description of the average or range of severity of dwarf mistletoe infection on the trees would be helpful. Also, why are Hawksworth DMR ratings not contained to just whole numbers in Fig. 4A? There seems to be levels of DMR that are at 0.5, which is not the standard DMR rating stated in the methods.

Specific comments:

Title and throughout MS: I’m not sure if the hyphen in dwarf mistletoe is needed. Literature tends not to use it so I recommend removing it. However, if the hyphen is going to be maintained, then just be consistent throughout MS as there are places that it is not present (e.g., ln 10).

Ln 67-69: Should be revised to: Because resin defenses have also been reported to be influenced by tree age, size, and growth rate [12,17], additional efforts to disentangle the relationships among these tree properties and mistletoe infection are needed.

Ln 110: Should revise to: and I collected one, 12-mm-diameter increment core

Ln 111: I’m not positive why the reference is here as it does not seem to relate to typical infection rates found in this area.

Ln 169: Revise to ‘Results and Discussion’

Ln 172: make ‘ducts’ singular

Ln 183: I’m curious about mean cross-sectional area of resin ducts vs total count of resin ducts. Would this be an appropriate way to test the idea of ‘more vs larger’ instead of total cross-sectional area?

Ln 185: Be consistent with italicized abbreviations, as other ‘i.e.’ were not italicized.

Ln 205: Revise second ‘pine’ to be singular

Ln 209: I think the single quote can be removed.

Lns 225-237: There is most likely an effect of severe dwarf mistletoe infection on calculating above-ground biomass using allometric equations. For example, some allometric equations have been shown (or are assumed) to not be appropriate for dwarf mistletoe infected trees (Shaw and Agne 2015, Botany 95: 231–246; Wanner and Tinnin 1986, Canadian Journal of Forest Research, 16(6): 1375-1378; Sala et al. 2001, Oecologia, 126(1): 42-52). This use of allometric equations for biomass of infected trees (especially severely infected/old) could be discussed.

Ln 239 and 355: Revise units for radial growth to be consistent throughout MS, and include units for BAI

Ln 246: Revise to DBH

Ln 252: Correct to P. aristata

Ln 283-286: A better description here and in the caption of what the numbers in Figure 5 are would be helpful for interpretation of the SEMs. Would standard errors of these estimates also help in interpretation?

Ln 292: Revise to mm2

Ln 320: italicize contorta
